# Waste Separation Behaviour of College Students under a Mandatory Policy in China: A Case Study of Zhengzhou City

**DOI:** 10.3390/ijerph17218190

**Published:** 2020-11-05

**Authors:** Mengge Hao, Dongyong Zhang, Stephen Morse

**Affiliations:** 1College of Information and Management Science, Henan Agricultural University, 15 Longzi Lake Campus, Zhengzhou East New District, Zhengzhou 450046, China; haomengge99@foxmail.com; 2Centre for Environment and Sustainability, University of Surrey, Guildford, Surrey GU2 7XH, UK; s.morse@surrey.ac.uk

**Keywords:** solid waste, waste separation, college students, influencing factors, China

## Abstract

The need for effective solid waste management (SWM) is an important environmental and public health issue. As a key way of minimizing municipal solid waste (MSW), source separation has in recent years become the centre of discussion in China. Following the example of Shanghai, the city of Zhengzhou introduced its mandatory waste separation measures on 1 December 2019. But does the mandatory regulation work? This study aims to investigate the waste separation behaviour of college students in Zhengzhou under the mandatory regulation and the motivations behind students’ behaviour. A questionnaire-based survey was carried out on 62 university campuses in Zhengzhou City, and a total of 1747 valid questionnaires were completed across these campuses and analysed. It was found that under Zhengzhou’s mandatory measures, college students do have a basic knowledge of waste separation and most are familiar with where kitchen waste should be placed, but they have problems categorizing some recyclables such as glass, hazardous waste such as lightbulbs and other waste such as cigarette butts and napkins. It was also found that college students’ waste separation behaviour, their attitude towards waste separation and the convenience of waste sorting facilities in Zhengzhou in the mandatory era have been improved compared to the era prior to mandatory waste separation. The results also indicate that most of college students (86.7%) always or sometimes undertake waste separation, and students majoring in science and senior year undergraduates are more likely to participate in the practice of waste separation. Other influencing factors of college students’ waste separation behaviour include convenience of waste sorting facilities, their willingness to separate waste, knowledge of a related field, attitude towards waste separation, peer pressure as well as the existence of a reward and penalty system. Management strategies for improving college students’ waste separation behaviour under mandatory regulation are also discusses and a number of recommendations for improvement are made.

## 1. Introduction

Improper and inefficient disposal of municipal solid waste (MSW) is often related to various environmental and public health problems [1,2]. China has been battling its rapidly increasing MSW in recent years. According to the Ministry of Housing and Urban-Rural Development of the People’s Republic of China [3], the quantity of MSW produced in China increased 7.3 times, from 31.32 million tonnes in 1980 to 228.2 million tonnes in 2018 while during the same period China’s GDP increased 73 times, from 191 to 13895 billion USD [4]. MSW source separation has been widely recognized as a means to minimize waste and enhance resource utilization [5]. A series of measures have been adopted by the central and local governments in China to promote MSW separation, but most of them have failed due to factors such as the lack of willingness of people, inadequate infrastructure and technology, poor coordination among the varied government institutions and weak enforcement [6]. In a government report in 2017, the Chinese premier Li Keqiang re-emphasized the importance of MSW separation and pledged to tackle the MSW problem in China. Some key cities were selected as pilots to implement the new MSW separation policy. Zhengzhou, as the capital of Henan Province in central China, was among these pilot cities.

Zhengzhou has been trying to promote waste classification in the city since October 1999 when it first introduced to its citizen the idea of recycling battery waste. In 2017, Zhengzhou welcomed its first local MSW policy which categorized solid waste into two types: recyclable and non-recyclable, and 18 pilot communities, representing about 1.4% of the population in the city, were selected to implement the policy. But the implementation of this policy was not successful, and the main reason was thought to be the lack of waste sorting infrastructure and the ineffective connection of waste source sorting, collection, and transportation [7].

Following Shanghai, which ushered in China’s mandatory waste separation era by introducing Shanghai Household Waste Management Regulation on July 1st, 2019, Zhengzhou issued Administrative Measures for the Classification of Municipal Solid Waste (hereinafter referred to as the Zhengzhou Measures) which took effect on December 1st, 2019. Since then waste separation has been incorporated into the legal framework in Zhengzhou. Zhengzhou Measures include six chapters and 39 articles, and according to the Measures, domestic solid waste is classified into four categories: recyclables, kitchen waste, hazardous waste, and other waste. Failing to comply with Zhengzhou Measures would result in a fine of 1000–3000 Yuan (about US$147–442) for entities and 50 Yuan (about US$7.37) for individuals. Entities and individuals that have done exceptionally well in waste separation would be rewarded. However, there is no clear indication regarding what is meant by ‘exceptional’ in this case. The responsible government departments in solid waste management (SWM) system are also stated in Zhengzhou Measures, including the City government, each District government, the Housing Security and Real Estate Management Department, Environmental Protection Department, Renewable Resource Recycling Management Department, etc.

To promote the implementation of Zhengzhou Measures, efforts were made by the Zhengzhou City government in improving waste sorting infrastructure and waste sorting related publicity. Currently, four types of outdoor rubbish bins labelled ‘recyclable waste’, ‘kitchen waste’, ‘hazardous waste’ and ‘other waste’ are placed on both sides of the road in Zhengzhou urban areas and the spacing between rubbish bins varies depending on the population density of the area. The distance between rubbish bins on busy commercial streets is 25–50 m, 50–80 m on arterial roads, and 80–100 m on other roads. For most residential communities in the city, three coloured rubbish bins are placed at the entrance of each unit of building blocks since December 2019, they are green, blue and yellow and labelled ‘kitchen waste’, ‘recyclables’ and ‘other waste’ respectively. But the labels are often not clear, and the waste from these bins is often mixed when collected and transported to its next destination. Because these bins are made in plastic material, they are easily damaged, and the replacement may not be in the same colour as the damaged one, which could confuse users regarding which bin to put their rubbish in. Since December 2019, rubbish bins of four colours with signs of ‘recyclable waste’, ‘kitchen waste’, ‘hazardous waste’ and ‘other waste’ have appeared at some university campuses in Zhengzhou City. But there is no clear indication that waste separation is mandatory on university campuses in the city.

However, the effective implementation of waste separation measures requires not only government regulation and adequate infrastructure but also the participation of every citizen [8], and knowing the factors that influence citizens’ participation rate is important in trying to improve their waste separation behaviour. As a group of citizens with knowledge and expertise, university students’ waste separation practice could, in theory, provide beacons (i.e., good examples) which others can learn from and follow [9]. College students are more likely to become decision makers and leaders of sustainability later in their career because at university they acquire technical and specialised knowledge necessary to make important decisions and to carry out legal, social, technological innovations for a more sustainable world [9,10,11]. In addition, involving the younger generation in environmental initiatives is a way of nurturing the contribution of conscientious citizens to environmentally sustainable SWM [12,13,14]. How college students deal with campus waste can be assumed to influence their participation in waste separation after their graduation, and affect their family’s waste separation behaviour, including subsequent generations [15]. Compared with other communities, colleges and universities arguably have a clearer and simpler SWM system [16]. And college students usually generate similar types of waste in certain concentrated key locations. For example, food waste, the highest component of waste generated on a Chinese university campus, is mainly generated in dining halls [16,17]. This should help the waste separation system on a university campus by making it easier to operate. Finally, college students are young and well-educated, which should make it easier for the communication of environmental sustainability practice [18]. Indeed, college campuses are almost ‘miniature cities’ in terms of infrastructure and governance, along with their young and well-educated ‘citizen’, and therein provide an ideal context for testing interventions designed to influence people’s behaviour [19,20], but little is known about the factors that influence the waste separation behaviour of college students under mandatory waste separation regulation, especially in countries like China which are undergoing extensive and rapid economic expansion and urbanization. Most of the studies on waste separation behaviour in China were conducted before the introduction of mandatory waste separation regulation in 2019 and few have looked at the factors that may influence waste separation behaviour after the introduction of the mandatory waste separation regulation. Also, research on the waste separation behaviour of students in the country is rare but for the reasons set out above this is an important group to consider. This study seeks to address these gaps in knowledge by exploring the factors which influence behaviour change of college students in waste separation in the era of mandatory waste separation in China.

The study has a number of limitations. First, the mandatory waste separation policy is still in its infancy, and there might be a delay when it comes to the implementation of a new regulation. The investigation at the heart of this study began one month after the policy was implemented, so the results can only reflect the reality at the beginning of the mandatory policy implementation. Further discussion on issues related to mandatory waste separation policy, such as how mandatory policy may change people’s attitudes towards waste separation and how to implement mandatory measures more effectively, can be analysed in future research. Second, waste separation behaviour is a complex human behaviour potentially affected by many factors. The factors included in this study are not all of the influences that may be involved. More influencing factors should be considered in future research. Third, this study focused on the impact of some influencing factors on behaviour under the mandatory waste separation policy. However, there may be other relationships among these variables, for example, the mandatory policy might have a direct influencing effect on people’s attitude towards waste separation and their willingness to separate waste. Finally, of course, the focus on college students means that the results cannot be readily extrapolated to the wider population in China. As noted above, college students have a set of characteristics that would distinguish them from the wider population.

## 2. Literature Review

Given the importance of waste separation in SWM, much empirical work has been done to explore participation rate in waste separation in China. By employing an intercept questionnaire-based survey of 287 undergraduate students in Zhengzhou two years before the mandatory waste separation regulation and using binary logistic regression analysis, Dai et al. [7] found that 10.1% of undergraduate students in Zhengzhou separated their waste in 2017. Zhang et al. [16] found that 7% college students separated three kinds of waste at source in Beijing. Wang et al. [21] found that 9% of residents always participated in waste separation in Hangzhou while Han and Yuan [22] found that 6.73% of residents have the habit of waste separation in Yangzhou. The reason for the low participation rate in waste separation, as Dai et al. [7] indicated, was the obvious lack of a suitable waste separation system in the city. However, Zhu [23] reported a participation rate of waste separation in Yaohua Street, Nanjing, China as high as 30–40%. Zhang and Wen [5] found that 23% of residents have source separated waste according to the waste separation method in Suzhou. The large variation in waste separation participation rate reported in the literature indicates there might be complex factors at play. Also, in order to increase people’s waste separation participation rate, identifying the various factors that could contribute to good waste separation behaviour has become a significant focus for the academic community.

Situational factors are widely discussed as influencing people’s waste separation behaviour. Situational factors (also called contextual factors and external conditions) refer to the environment that resident individuals face when performing environmental behaviour choices, including limited time, space, and infrastructure, how it is managed, economic measures and people’s trust in the SWM system (i.e., how strongly and fairly the regulation is enforced) [24,25,26]. Some of the contextual factors may facilitate or constrain pro-environmental behaviour and influence individual motivation [27,28]. Zhang et al. [25] concluded that situational factors significantly predicted household waste separation behaviour when looking at a case in Guangzhou, China. The effect of economic measures on encouraging pro-environmental behaviour has been confirmed by many scholars. For example, Fan et al. [24] found that, based on samples from China and Singapore, the higher the benefits people perceived from doing waste separation, the more they are likely to separate waste. Some studies found that the levels of trust people have in the enforcement of waste separation regulation has an important impact on their waste separation behaviour [5]. For instance, Zhang and Wen [5] found that when residents do not trust the waste separation system (i.e., they discovered that the waste they separated in different bins was mixed together when it was collected and transported) they would not undertake household waste source separation. By conducting in-depth interviews with 323 residents in China, Chen et al. [29] also found that the lack of sorting facilities and the incompleteness of the entire recycling system are the reasons why most are not willing to do waste separation. The survey conducted by Ghani et al. [30] demonstrated that the public tend to participate more in food waste separation at source when opportunities and facilities are adequately prepared. They also found that good situational factors such as storage convenience and collection times can enhance the extent of public participation in food waste separation in Malaysia [30].

Waste separation behaviour, as a type of individual behaviour, is also affected by people’s individual motivation (such as attitude towards waste separation, willingness to separate waste, subjective norms, environmental awareness and knowledge). Dai et al. [7] found a negative connection between attitude towards waste separation and waste separation behaviour in Zhengzhou, China. Some studies found that attitude towards waste separation is the main predictor regarding willingness to undertake waste separation and based on this it is possible to predict people’s actual waste separation behaviour [24,25]. Some found that subjective norms positively affect people’s willingness to separate waste and their separation behaviour [16]. Combined with the theory of planned behaviour, Zhang et al. [25] conducted a case study in Guangzhou, China, and found that moral judgment and ‘felt obligations’ are also identified as key variables that affect individual behaviour. Wang et al. [31] found that ecological civilization education and related knowledge are statistically related to college students’ waste separation behaviour in China. Vicente-Molina et al. [10] used data from 2226 respondents spanning four countries (United States, Spain, Mexico, and Brazil) to study the influence of college students’ environmental knowledge on pro-environmental behaviour and found that environmental knowledge significantly and positively influenced pro-environmental behaviour. Some studies found that recyclers in general are more knowledgeable of recycling issues than non-recyclers [32,33]. However, Mousavi et al. [34] found no significant correlation between environmental knowledge and waste recycling behaviour.

Finally, common demographic factors i.e., gender, age, educational level and income have also been considered to be influencing factors of people’s waste separation behaviour. Fan et al. [24] performed a comparative study to explore the similarities and differences between the determinants of waste separation behaviour in Shanghai and Singapore by surveying 1100 urban residents from these two places, and found that people’s age, education and income were significantly positively related to their waste separation behaviour in these two places. Matsumoto [35] found that women and older people in Japan were more likely to participate in waste recycling activities when looking at the curbside recycling programs. However, the results of such studies have often been contradictory. For example, Ghani et al. [30] found that gender was not a statistically significant influencing factor for the participation in source separation of food waste for households in Malaysia, and Botetzagias et al. [36] found that demographic factors did not have statistically significant influence on people’s recycling willingness in Greece.

Based on the influencing factors of pro-environmental behaviour and to facilitate the implementation of waste management policies, various policy instruments have been applied around the world, including legal, economic and social measures [1]. Legal measures are measures within the field of direct influence of the government and aim to direct or forbid practices that impact on public health and environment. Economic measures provide economic incentives and disincentives for specific actions and behaviour. Social measures seek to inform, interact and engage with stakeholders to improve the users’ behaviour [1]. Steg and Vlek [37] and Fan et al. [24] defined legal and economic measures as structural measures which aim to change contextual factors, and social measures as informational measures which intended to target the motivational factors mentioned above.

Three types of structural measures are commonly used. First, improving the availability and convenience of infrastructure, such as the number and/or location of litter bins, has been mentioned by some researchers as essential in promoting waste sorting behaviour [38,39]. Second, legal measures have been proposed by some scholars as means to improve people’s waste separation behaviour. For example, Nguyen et al. [40] recommended the creation of laws and regulations and follow through on their enforcement to facilitate effective waste separation at the household level and thereby increase the trust of residents in government’s waste separation management. Third, temporary taxes or subsidies can be imposed to users to increase a green consumption [41]. Based on China’s experience, Fan et al. [24] pointed to the importance of the role of subsidies in improving residents’ pro-environmental behaviours. Indeed, the participation rate of waste separation reached 30–40% in Yaohua Street, Nanjing, China by providing rewards to encourage residents to participate in waste separation [23]. Given the development of online payment in China, Fan et al. [24] suggested that the government can collaborate with Alipay (a popular mobile online payment application in China developed by Alibaba Group) and turn green account scores into a direct shopping discount.

Informational/social measures aim to change people’s perceptions and motivations without changing the external context in which choices are made [37]. Pro-environmental prompts that aim to increase environmental awareness and knowledge are effective in improving pro-environmental behaviours [42,43]. Based on the theory of planned behaviour and the norm-activation theory, moral motivations (such as environmental values, ascription of responsibility, personal norms, social norms, etc.) can motivate people to participate in environmental behaviours [44,45]. The sense of responsibility for environmental protection can be strengthened by setting role models [46]. Zhang et al. [16] proposed that publicity should emphasize the adverse consequence of environmental pollution caused by solid waste. Fan et al. [24] believed that having everyone’s sorting behaviour publicly on display by using scores to allow for comparison is an effective method to encourage good sorting behaviour and discourage bad sorting behaviour. Chan and Bishop [47] recommended that the government can encourage inhabitants to participate in charity activity and commend their behaviour publicly.

Based on above literature review, this study intends to explore waste separation behaviour and its influencing factors under the mandatory waste separation policy based on a survey of college student in Zhengzhou. A model is presented as Figure 1 to explain the relations of variables under investigation.

## 3. Methodology

After a pilot study in November 2019, a semi-structured questionnaire was finalized and comprised a mix of closed and open-ended questions aimed to uncover respondent participation in waste separation and the factors which may influence their behaviour under mandatory measures. The questionnaire comprised 23 questions set out in three sections. The first section covered demographic information (gender, year of study, etc.). The second section comprised four questions which aimed to explore the respondents’ waste source separation behaviour, and the condition of practicing/not practicing waste separation. The third section comprised 14 questions, which sought to explore the influencing factors of waste separation behaviour, such as respondents’ attitudes towards waste separation, their willingness to separate waste and understanding of relevant knowledge. For some of the questions the respondents were free to select more than one answer. The inclusion of most of the variables were based on existing research (see Table 1), but a few variables were tailored specifically for this research, for example, conditions under which people would do/not do waste separation, effect of distance to the bin, reward, penalty, time constraint on the waste sorting behaviour (marked ‘-’ in Table 1).

After obtaining ethical approval from the College of Information and Management Science, Henan Agricultural University, the full survey was undertaken in January 2020. A group of one postgraduate and five undergraduate students was employed to conduct the data collection, and they received training in survey skills before the fieldwork.

Face-to-face intercept interviews were carried out on 59 university campuses in Zhengzhou. Six investigators, two as a group, visited 1–2 universities a day and met students in the libraries, classrooms, dining halls, dormitories and corners in the buildings, out-door sports areas, and asked them to complete the questionnaire and return them face to face. The questionnaire was often handed out in batches so the students can work on them at the same time. From 1 to 15 January 2020, a total of 2000 questionnaires were distributed, of which 1689 were considered valid. After the 15 January when most students went home for the Chinese New Year, the questionnaire was sent to potential respondents studying in another three universities (there were 62 universities in total in Zhengzhou at the time of the survey), and a total of 58 completed questionnaires were returned. This makes a total response of 1747 (1689 + 58) completed questionnaires. There was no reward for returning a questionnaire (online or offline).

SPSS 24.0 (IBM, Armonk, NY, USA) was employed to store and analyse the data. Crosstab analysis, Chi-square test and Fisher’s exact test were used to explore the factors affecting the waste separation behaviour of college students under the mandatory waste separation regulation.

## 4. Results and Discussion

### 4.1. Sample Description

Of the total respondents, 42.7% were male and 57.3% were female, and 96.1% were undergraduates while 3.9% were doing a postgraduate programme (including masters and PhDs) (which is around the average level of postgraduate enrolment rate in universities in Zhengzhou). In terms of subject area, 46% were studying an arts-based subject while 54% were studying a science-based subject. Within the 1679 undergraduate students, 1672 indicated their year of study and of these 31.7% were in their first year of study, 29.4% were in their second year, 28.1% in their third year and 10.8% in their final year. Most postgraduate students in China receive financial support from their tutors, some from their universities or/and national funding institutions while some have part-time jobs. Most undergraduate income comes from their parents. Hence postgraduate total income and undergraduate income from their parents were considered to be their disposable income. In the total sample of 1747 students, 1642 indicated their disposable income and 3.2% had a monthly disposable income of less than 600 Yuan, 37.9% had between 600 and 1000 Yuan, around half (53.3%) between 1000 and 2000 Yuan, and 5.6% had more than 2000 Yuan.

### 4.2. College Students’ Waste Separation Behaviour under the Mandatory Measures

To understand Chinese college students’ waste separation behaviour, a single-choice question was asked: ‘do you participate in waste separation?’ 232 (13.3%) respondents said ‘never’ (Table 2), 311 (17.8%) said ‘always’, while more than half (68.9%) said ‘sometimes’. The waste separation participation rate is used as an indicator for the waste separation behaviour, the higher the participate rate the better the behaviour. Clearly, Zhengzhou college students’ waste separation behaviour has improved from before the mandatory era (10.1%, [7]) to 17.8%. The mandatory Zhengzhou Measures have obviously served this purpose as reported by this study.

To understand the rationale behind students’ behaviour, two multiple-choice questions were asked to 1204 respondents who reported that they sometimes separated waste and the answers are shown in Figure 2. The results suggest two factors that matter the most when Zhengzhou’s college students decide whether or not they would participate in waste separation, and these were how easy it is to access rubbish bins (28.1% would do if it is easy and 45.5% would not do if it is not easy) and how much trouble (27.1% would do if less trouble and 31.5% would not do if more trouble) is involved when separating the waste. Constraints on time, strict regulation, peer pressure, trust in the SWM system and publicity are also important factors that college students consider when making waste separation choices.

In order to understand the driving factors of students’ good waste separation behaviour, in-depth interviews were conducted with students who reported to always separate waste. The benefits of waste separation for environmental protection are the most frequently mentioned factor when they were asked why they would participate in waste separation. Other factors such as the convenience of waste sorting facilities, peer pressure and the strictness of relevant university regulation and publicity of waste separation issues were also mentioned.

The above factors were also confirmed by 232 respondents who reported to never separate waste in in-depth interviews. Clearly, apart from convenience of waste sorting facilities, how much trouble is involved when doing waste separation, time constraint, the benefits of waste separation for environmental protection, the respondent’s knowledge of waste separation, student’s trust in SWM system, peer pressure, strictness of relevant university regulation plays important roles in college students’ waste separation behaviour. This raised the question as to what was mentioned earlier: why does waste separation is not mandated on the university campuses in Zhengzhou? As one of the members of the Environmental Protection Association at one of the universities in Zhengzhou indicated that there are various reasons of not mandating waste separation on campus in Zhengzhou, the first and the foremost is that the universities do not want to do it due to the potential (huge) effort involved and the possible failure if waste separation were mandated on campus. Second, although some universities in other areas of China, such as Shanghai and Guangzhou, have started mandatory campus waste separation, there has not to date been any successful stories reported in the literature. Finally, teaching and providing guidance, instead of mandating, is the fundamental principle of an educational institute.

### 4.3. Influencing Factors of College Students’ Waste Separation Behaviour

#### 4.3.1. Demographic Factors

Chi-square tests between respondents’ demographic information and their waste separation behaviour were performed and the results are shown in Table 3.

Pearson Chi-square test results in Table 3 indicate that whether or not a college student would participate in waste separation was not statistically related to their gender, the level of programmes (postgraduate or undergraduate) they were following and their disposable income.

This finding echoes the result found by Fan et al. [24] that female’s behaviour in household waste separation is not significantly different from male’s in Shanghai. The reason for this result, as Fan et al. [24] indicated, may be related to males and females tending to share household chores.

However, college students’ behaviour in waste separation was statistically related to the area of their study (*p* < 0.05), and students taking Science subjects tended to do more waste separation than those taking Arts. This result is consistent with the result from Wang et al. [31] who studied the factors influencing Chinese college students’ waste separation behaviour by surveying students in 152 universities of 30 regions in China. The reason could be the curriculum design in Chinese university courses as environmental protection and resource reservation etc. tend to be covered more in the science-based subjects.

Table 3 indicates that undergraduate college students’ behaviour in waste separation was closely related to their year of study (*p* < 0.05). A crosstab analysis in the same table indicates that senior year students were more likely to participate in waste separation than their juniors. This result is consistent with the finding of Dai et al. [7] who investigated the factors influencing people’s willingness and behaviour in waste separation by surveying 285 college students and 330 residents in Zhengzhou in 2017 (2 years before the city started the mandatory waste separation regulation) and found that older people were more likely to recycle. The finding is also in line with the results of Sidique et al. [50], Saphores et al. [51] and Meneses and Palacio [52].

#### 4.3.2. Convenience of Waste Sorting Facilities

To understand whether the convenience of waste sorting facilities influences respondent’s waste separation behaviour, respondents were asked to agree or disagree with two provided statements (as shown in Table 4).

Table 4 indicates that a majority of respondents (91.9%) would do waste separation if there is a bin around and if there were clear signs on it, but only 55.9% of them believed that there were enough bins with clear signs that were easy to reach on their campus. Although this number (55.9%) is still smaller than expected, the convenience of waste sorting facilities has been greatly improved relative to the prior mandatory era (17.0%, [7]).

Chi-square test results indicate that the infrastructure of waste separation has an important impact on waste separation behaviour of college students (*p* < 0.01 and *p* < 0.05). The more convenient the waste separation infrastructure is then the higher the participation rate of waste separation. This echoes Wang et al. [21] and Liao and Li’s [48] conclusion that it is important to make the waste separation easier by improving the infrastructure. But the fact that only a little more than half of the respondents considered their campus had sufficient waste sorting facilities indicates that the poor waste separation infrastructure has been one of the problems contributing to the low level of waste separation behaviour on university campuses in Zhengzhou.

#### 4.3.3. Willingness to Separate Waste

To assess whether respondent’s willingness to separate waste influences their waste separation behaviour, respondents were asked to agree or disagree with a few statements that described their willingness to separate waste (as shown in Table 5). Respondents who choose to agree with the statements were considered as being willing to separate waste.

Table 5 indicates that 86.3% of respondents wanted to separate waste at source. However, when they realized that they had to overcome some difficulties to do the waste separation (i.e., walking some distance or having a tight schedule), their willingness dropped significantly. Some 74.2% would walk some distance to put waste into the correct bin and 63.8% would squeeze time out of their schedule to do the waste sorting.

Chi-square test results indicate that the willingness of college students to separate waste in Zhengzhou is statistically related to their waste separation behaviour (*p* < 0.01); the stronger the willingness, the higher the participation rate in waste separation. This finding is consistent with the result from Zhang et al. [25] who conducted a case study in Guangzhou to explore the influencing factors on waste separation behaviour by using structural equation modelling. It also agrees with the result of Meng [53].

#### 4.3.4. Knowledge in Waste Separation

To assess the respondents’ knowledge in waste separation, they were asked how familiar they were with standards relating to waste separation in Zhengzhou (Table 6).

Table 6 indicates that only 5.5% of the respondents were familiar with Zhengzhou’s waste separation standards, indicating college students’ relatively low levels of knowledge in waste separation.

Even though the overall level of knowledge in waste separation is low, the results indicate a significant relationship between college students’ knowledge of waste separation and their waste separation behaviour (*p* < 0.01); the better the knowledge the better the behaviour, a result which agrees with those of Meng [53] and Qu et al. [54]. This result, along with the overall low levels of waste separation knowledge among college students, highlights the importance of enhancing knowledge in SWM in order to improve people’s waste separation behaviour. Hence more effort needs be made by the local government and the university to promote the idea of waste separation, to raise awareness and to reinforce the implementation of Zhengzhou Measures among college students.

To further understand how equipped the students are with waste separation related knowledge, they were provided with a list of different kinds of waste and asked to put them in the right categories based on what is stated in Zhengzhou Measures (i.e., recyclable waste, kitchen waste, hazardous waste and other waste). Figure 3 presents the percentage of people who did it correctly.

Of the 1730 people who responded to this question, more than 70% could correctly separate textiles, tea leaves, floor sweepings, fruit residues, medicines, batteries, fish bone and paperboard. This implies that most college students in Zhengzhou have the basic knowledge of where the common waste should be placed. However, it is surprising that some make mistakes when categorizing common recyclables such as glass (with an accuracy rate of 45.4%), hazardous waste such as light tubes (with an accuracy rate of 42.1%), and other waste such as cigarette ends (with an accuracy rate of 36.3%) and napkins (with an accuracy rate of 29.5%).

The reasons for failing to correctly categorize these wastes are multiple. The first, as one of the interviewees pointed out, is the confusing categorization for the waste of items made of even the same material. For example, glass is in the category of ‘recyclable waste’ if it is intact i.e., glass bottles and jars, but ‘other waste’ if it is broken. The second reason is that some respondents based their judgement purely on where garbage was generated. For instance, napkins are mostly generated during meals so they are considered as ‘kitchen waste’ when in fact they are in the ‘other waste’ category. Finally, some respondents related the waste category to what they have known about the waste itself. For example, cigarette ends were regarded as hazardous waste by many because of a popular slogan ‘Smoking is harmful to health’ when they should be in the ‘other waste’ category. There is need to raise awareness of these categories and based upon the findings of the research special attention should be paid to napkins, cigarette ends, light tubes, and glass when distinguishing hazardous waste, recyclables and other waste.

#### 4.3.5. Attitude towards Waste Separation

To understand whether respondent’s attitude towards waste separation influences their waste separation behaviour, they were asked to agree or disagree with a few statements that describe the benefit of waste separation (as shown in Table 7). Respondents who choose to agree with the statements are considered as having a positive attitude towards waste separation.

Table 7 indicates that 92.8% of college students agreed that waste separation can improve our living environment and public health, 90% of them thought universities should act as role models in waste separation, and 84.3% believed that waste separation would reduce our carbon footprints on the earth and it was good for the environment.

The results of Fisher’s exact test indicate that the attitude of college students in Zhengzhou towards waste separation is related to their waste separation behaviour (*p* < 0.01). Those who had a positive attitude towards waste separation did practice waste separation better. This finding is partly in line with the result of Zhang et al. [16] who surveyed college students in Beijing before the city started the mandatory waste separation regulation and found a positive correlation between their attitude towards waste separation and their behaviour. However, in the same study, Zhang et al. [16] also found that students in Beijing who had a positive attitude towards waste separation did not necessarily have better waste separation behaviour. The differences in the results might be due to regional differences, but the regulatory influence (whether or not a mandatory regulation is in place) may also play an important role.

#### 4.3.6. Subjective Norms

It is known that subjective norms reflect the individual’s impression of others’ feelings about performing the behaviour [55]. Subjective norms are essentially peer pressure [56]. If an individual perceives that others who are important to them (such as family, friends and role models) have a positive view of waste separation, then an individual is likely to do waste separation. As a potential influencing factor, subjective norms are also evaluated in this study by asking respondents to agree or disagree with provided statements (as shown in Table 8).

It can be seen from Table 8 that the majority of college students sampled (88%) would participate in waste separation if people around them also engage in it, which indicates the prevailing influence of peer pressure in waste separation behaviour among Chinese college students. But only 65.3% would feel guilty if they did not separate their waste before dumping it into the bin or if they did not do it properly.

Chi-square test results indicate that the impact of subjective norms on waste separation behaviour of college students in Zhengzhou is statistically significant (*p* < 0.01). This result agrees with Zhang et al.’s [16] result that 91% of college students in Beijing would definitely do waste separation if people around them did it.

#### 4.3.7. Reward and Penalty System

To determine whether a reward and penalty system would affect respondents’ behaviour in waste separation, two reward and penalty related scenarios were presented to the respondents (as shown in Table 9).

Table 9 indicates that the majority of respondents (92.4%) would separate waste when it is assumed that there would be a penalty for not participating in waste separation, while more respondents (94.3%) would do it when it is assumed that there would be a reward for participating in waste separation. However, 79.5% of respondents would do waste separation regardless of whether or not there were reward and penalty measures in place.

Chi-square test results indicate that the reward and penalty system is statistically related to respondents’ waste separation behaviour (*p* < 0.01). People with better waste separation behaviour are more sensitive to reward and no reward/penalty at all, and less sensitive to penalty measures. Clearly ‘carrot’ works better than ‘stick’ when it comes to waste separation. Indeed, a good reward system could play an important role in improving people’s waste separation behaviour. In fact, successful examples have been reported by Zhu [23] in Yaohua, a sub-district of Nanjing City, where a ‘waste sorting reward platform’ was established and residents who participated in waste separation would receive points and gifts. However, although fewer, some people obviously see waste sorting as their duty and giving a reward/penalty would be regarded by them as a demotion to their good waste separation behaviour and no longer want to carry it out. This result is consistent with the findings of Berglund [57] and Dahlén and Lagerkvist [58]. Hence, the government has to be careful when implementing a rewarding system as over-doing it might discourage people with existing good waste sorting behaviour.

## 5. Conclusions and Recommendations

This paper employed a semi-structured questionnaire survey of college students from 62 universities in Zhengzhou to gather information on the waste separation behaviour of this group and the influences at play after the implementation of mandatory waste separation measures in the city. The results indicate that although 92.8% of the respondents agree that waste source separation is good for the environment and public health, only 17.8% of them always participated in waste separation practices. Most of them (68.9%) sometimes (depending on the situation) participated in waste separation, and 13.3% of them never did it. Although respondents have basic knowledge of waste separation and kitchen waste (i.e., tea leaves, fruit residues, and fish bone) is what they are most familiar with, they find it difficult to identify some common MSW, such as glass, light tubes, cigarette ends, and napkins.

The participation rate of college students in waste separation before and after the implementation of the mandatory measures in Zhengzhou increased from 10.1% to 17.8%, indicating an improvement in waste separation behaviour among this group. Although the waste separation participation rate of college students in Zhengzhou is still low, this result suggests that the mandatory waste separation measures do work.

The waste separation behaviour of college students in Zhengzhou was not found to be statistically related to their gender, the level of programmes (postgraduate or undergraduate) they take and their disposable income, but is related to their subject areas and year of study (undergraduates only). Students in Science tend to do waste separation more than those in Arts. For undergraduates, senior year students are more likely to participate in waste separation than junior students.

Apart from the demographic factors, other factors were also found to affect college students’ waste separation behaviour significantly. These included contextual factors i.e., convenience of waste sorting facilities, reward and penalty system, and motivational factors i.e., willingness to separate waste, subjective norms, knowledge in related field, attitude towards waste separation. All these factors are found to be significantly and positively related to college students’ waste separation behaviour under mandatory waste separation measures.

However, people have mixed feelings to reward and penalty system when it comes to waste separation. While a good reward system is preferred, a penalty is not favoured by many, but some people consider waste separation as their duty and would rather do it without any reward.

Although there is still room to improve, the convenience of waste sorting facilities (increased from 17.0% prior to the mandatory regulation to 55.9% at the time of the study) and wider publicity of waste separation after the introduction of the mandatory measures has helped generate a positive attitude towards waste separation into good waste separation behaviour. Compared to the prior mandatory waste separation era, the attitude of college student towards waste separation in Zhengzhou improved from 87.0% to 92.8% in the mandatory era.

Based on the above conclusion, the following measures for improving college students’ waste separation behaviour are suggested: Firstly, the universities and local governments should do more work to improve their SWM system. People need to have confidence that any effort they make to separate waste is not negated during the collection and transportation phase, so a complete and trustworthy SWM system would lay the groundwork for the success of waste separation at source. The authority could also invest more in waste sorting facilities to make waste sorting easier. There needs to be adequate provision of disposal bins along with clear classification signs on dustbins so people can do the correct sorting. On the other hand, intelligent waste sorting facilities could be introduced to the campus/community, intelligent separation techniques (such as image processing technology, automatic sorting technology) could help optimize the separation process and increase classification accuracy rate.

Secondly, instead of penalizing people for not separating their waste, the Zhengzhou City government should make more effort to support people for good waste sorting behaviour, and this reward system has to be carefully tailored so it would not hurt the feelings of people who consider waste sorting as their duty. At the same time, the local government or the community could unveil the outcomes of waste separation supervision from time to time to prevent citizens from thinking that their participation in waste separation is ineffective.

In addition, knowledge in the field of waste separation should be publicized further within college students. Not only the harmful effects of solid waste pollution, but also the benefits of waste separation for the global environment and public health should be publicized. It is crucial to get people familiar with the detailed information about waste separation, i.e., how many categories are there and what belongs to which category.

Furthermore, social norms, nudges and feedback could be designed to help college students to develop a habit of waste separation and do their waste separation better in the long term. For example, waste separation behaviour could be regarded as an indicator and incorporated into the index system related to the selection activity of students’ personal honour and scholarship.

Last but not least, more attention should be paid to students in Arts and in their junior year when promoting campus waste separation. Apart from the general promoting strategies mentioned above, it might be a good idea to revise curriculum design in Chinese universities and add some courses that are related to environmental protection and resource reservation in Arts. It may also be useful if waste sorting campaigns are held more often for all first-year undergraduate students.

It does need to be stressed here that college students may not necessarily be representative of the wider Chinese population and thus some care needs to be taken when extrapolating the points made above. Nonetheless some of these points, such as supporting good waste behaviour, could have a wider resonance and future research which compares the use of penalties versus inducements is required.

## Figures and Tables

**Figure 1 ijerph-17-08190-f001:**
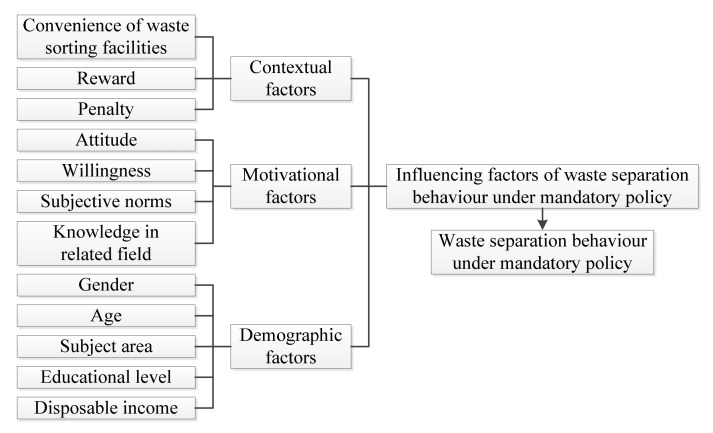
The model which underpins the research.

**Figure 2 ijerph-17-08190-f002:**
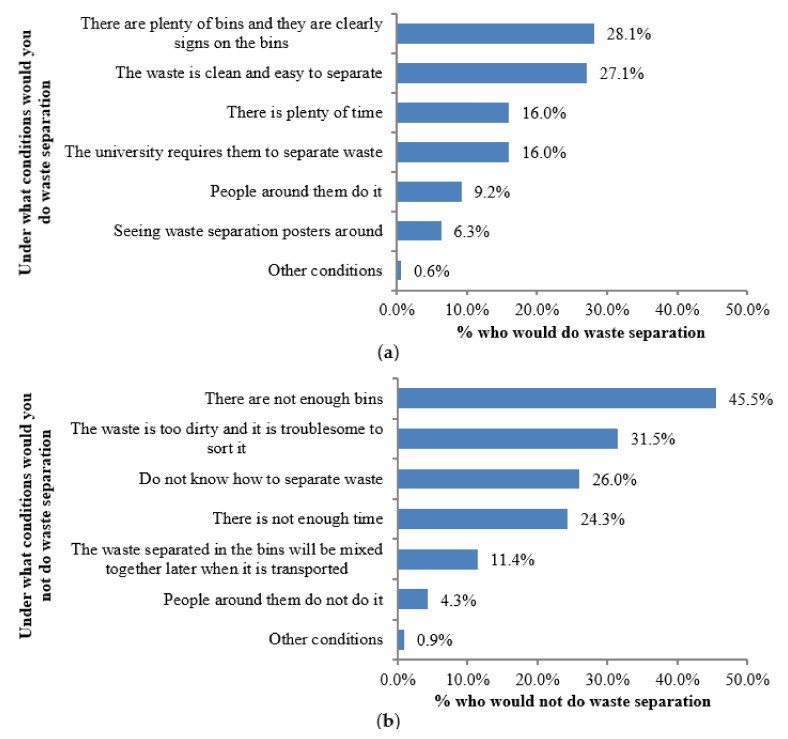
Conditions under which respondents would do/not do waste separation (N = 1204). (**a**) Conditions under which respondents would do waste separation. (**b**) Conditions under which respondents would not do waste separation.

**Figure 3 ijerph-17-08190-f003:**
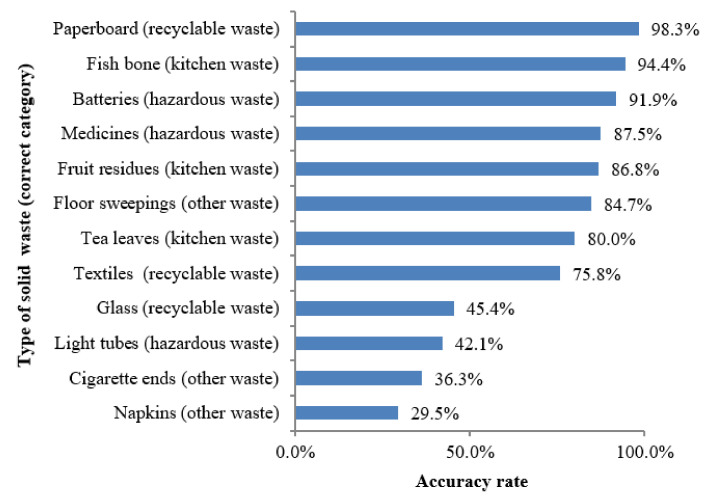
The accuracy rate of waste categorizing.

**Table 1 ijerph-17-08190-t001:** Sources of variables and questions in the questionnaire.

Variable	Question	Sources
Waste separation behaviour	Do you participate in waste separation?	Wang et al. (2020) [31]
Conditions of doing/not doing waste separation	Under what conditions would you do/not do waste separation?	-
Reasons for not participating in waste separation	What are the reasons why you not do waste separation?	Zhang and Wen (2014) [5]; Zhang et al. (2017) [16]
Convenience of facilities	I would do waste separation if there is a bin around and if there are clear signs on it.	-
	There are enough bins with clear classification signs on our campus and they are easy to reach.	Zhang et al. (2017) [16]; Liao and Li (2019) [48]
Reward and penalty system	Without any reward and penalty measures, would you do waste separation?	Zhang et al. (2015) [25]
	If you would be penalized for not separating your garbage, would you do waste separation?	-
	If you would be rewarded for waste separating, would you do waste separation?	-
Attitude	Waste separation can improve our living environment and public health.	Bortoleto et al. (2012) [49]; Zhang et al. (2015) [25]
	Universities should act as role models in waste separation.	Zhang et al. (2017) [16]
	Waste separation helps reduce our carbon footprints and is good for the environment.	Nguyen et al. (2015) [40]
Willingness	I want to separate waste at source.	Zhang et al. (2017) [16]
	I will walk some distance to put waste into the right bin.	Zhang and Wen (2014) [5]; Zhang et al. (2017) [16]
	Even if I have a very tight schedule, I would still try to find a way to do garbage sorting.	-
Subjective norms	If people around me do waste separation, I would do it too.	-
	I feel guilty if I do not separate my waste or if I do not do it properly.	Nguyen et al. (2015) [40]; Zhang et al. (2017) [16]
Knowledge in related field	Are you familiar with the waste separation standards in Zhengzhou?	Zhang et al. (2017) [16]
Disposable income	How much do you have for living expenses in a month?	-

- questions specifically designed for this study.

**Table 2 ijerph-17-08190-t002:** College students’ current waste separation behaviour.

Question: Do you Participate in Waste Separation?	Frequency	Percentage (%, N = 1747)
Never	232	13.3
Sometimes	1204	68.9
Always	311	17.8

**Table 3 ijerph-17-08190-t003:** Demographic information and waste separation behaviour.

Demographic Factors	Categories	Waste Separation Behaviour Top Figures: Observed (expected) Counts Bottom Figures: % within Column			Total	Chi-Square Tests Figures in Brackets: Degree of Freedom
		Never	Sometimes	Always		
Gender	Male	106 (99.1) 45.7	499 (514.1) 41.4	141 (132.8) 45.3	746 42.7	Pearson Chi-Square: 2.507 (2) ns Likelihood Ratio: 2.500 (2) ns Linear-by-Linear Association: 0.012 (1) ns N of Valid Cases: 1747
	Female	126 (132.9) 54.3	705 (689.9) 58.6	170 (178.2) 54.7	1001 57.3	
Programme	Undergraduate	224 (223) 96.6	1158 (1157.1) 96.2	297 (298.9) 95.5	1679 96.1	Pearson Chi-Square: 0.447 (2) ns Likelihood Ratio: 0.438 (2) ns Linear-by-Linear Association: 0.424 (1) ns N of Valid Cases: 1747
	Postgraduate	8 (9) 3.4	46 (46.9) 3.8	14 (12.1) 4.5	68 3.9	
Subject area	Arts	114 (102.9) 50.9	552 (545.9) 46.5	123 (140.2) 40.3	789 46.0	Pearson Chi-Square: 6.212 (2) ** Likelihood Ratio: 6.236 (2) ** Linear-by-Linear Association: 6.103 (1) ** N of Valid Cases: 1717
	Science	110 (121.1) 49.1	636 (642.1) 53.5	182 (164.8) 59.7	928 54.0	
Year of study (undergraduates only)	1st year	75 (71.0) 33.5	378 (360.2) 32.7	77 (91.0) 26.4	530 31.7	Pearson Chi-Square: 14.173 (6) ** Likelihood Ratio: 15.217 (6) ** Linear-by-Linear Association: 6.736 (1) *** N of Valid Cases: 1672
	2nd year	65 (65.9) 29.0	344 (336.7) 29.8	83 (85.1) 28.4	492 29.4	
	3rd year	71 (62.8) 31.7	304 (324.3) 26.3	94 (81.9) 32.2	469 28.1	
	4th year	13 (24.2) 5.8	130 (134.8) 11.2	38 (34.1) 13.0	181 10.8	
Disposable income	Within 600 Yuan	11 (7.2) 4.9	30 (36.2) 2.7	12 (9.6) 4.1	53 3.2	Pearson Chi-Square: 9.444 (6) ns Likelihood Ratio: 9.163 (6) ns Linear-by-Linear Association: 0.331 (1) ns N of Valid Cases: 1642
	600–1000 Yuan	78 (84.5) 35.0	446 (425.4) 39.7	98 (112.1) 33.1	622 37.9	
	1000–2000 Yuan	118 (119) 52.9	590 (599.1) 52.5	168 (157.9) 56.8	876 53.3	
	Above 2000 Yuan	16 (12.4) 7.2	57 (62.2) 5.1	18 (16.4) 6.0	91 5.6	

ns = not significant at *p* > 0.1; ** Statistically significant at *p* < 0.05; *** Statistically significant at *p* < 0.01.

**Table 4 ijerph-17-08190-t004:** Convenience of waste sorting facilities and waste separation behaviour.

Convenience of Facilities	Waste Separation Behaviour Top figures: Observed (expected) Counts Bottom Figures: % within Column			Total	Chi-square Tests Figures in Brackets: Degree of Freedom
	Never	Sometimes	Always		
Statement 1: I would do waste separation if there is a bin around and if there are clear signs on it					
Agree	191 (213.1) 82.3	1118 (1106.1) 92.9	296 (285.7) 95.2	1605 91.9	Pearson Chi-Square: 48.595 (4) *** Likelihood Ratio: 36.186 (4) *** Linear-by-Linear Association: 32.229 (1) *** N of Valid Cases: 1747 Note: 1 cell (11.1%) have expected count less than 5.
Disagree	17 (4.8) 7.3	16 (24.8) 1.3	3 (6.4) 1.0	36 2.1	
Not sure	24 (14.1) 10.4	70 (73.1) 5.8	12 (18.9) 3.8	106 6.0	
Statement 2: There are enough bins with clear classification signs on our campus and they are easy to reach					
Agree	110 (129.6) 47.4	677 (672.6) 56.2	189 (173.7) 60.8	976 55.9	Pearson Chi-Square: 11.211 (4) ** Likelihood Ratio: 10.998 (4) ** Linear-by-Linear Association: 8.830 (1) *** N of Valid Cases: 1747
Disagree	43 (32.0) 18.5	159 (166.1) 13.2	39 (42.9) 12.5	241 13.8	
Not sure	79 (70.4) 34.1	368 (365.3) 30.6	83 (94.4) 26.7	530 30.3	

** Statistically significant at *p* < 0.05; *** Statistically significant at *p* < 0.01.

**Table 5 ijerph-17-08190-t005:** Willingness to separate waste and waste separation behaviour.

Willingness	Waste Separation Behaviour Top Figures: Observed (expected) Counts Bottom Figures: % within Column			Total	Chi-Square Tests Figures in Brackets: Degree of Freedom
	Never	Sometimes	Always		
Statement 1: I want to separate waste at source					
Agree	177 (200.3) 76.3	1041 (1039.3) 86.4	290 (268.5) 93.3	1508 86.3	Pearson Chi-Square: 34.697 (4) *** Likelihood Ratio: 33.796 (4) *** Linear-by-Linear Association: 29.281 (1) *** N of Valid Cases: 1747
Disagree	15 (7.0) 6.5	32 (36.5) 2.7	6 (9.4) 1.9	53 3.0	
Not sure	40 (24.7) 17.2	131 (128.2) 10.9	15 (33.1) 4.8	186 10.7	
Statement 2: I will walk some distance to put waste into the right bin					
Agree	142 (172.1) 61.2	898 (893.2) 74.6	256 (230.7) 82.3	1296 74.2	Pearson Chi-Square: 47.126 (4) *** Likelihood Ratio: 41.028 (4) *** Linear-by-Linear Association: 36.179 (1) *** N of Valid Cases: 1747
Disagree	33 (13.9) 14.2	60 (72.4) 5.0	12 (18.7) 3.9	105 6.0	
Not sure	57 (45.9) 24.6	246 (238.5) 20.4	43 (61.6) 13.8	346 19.8	
Statement 3: Even if I have a very tight schedule, I would still try to find a way to do garbage sorting					
Agree	115 (147.9) 49.6	777 (767.7) 64.5	222 (198.3) 71.4	1114 63.8	Pearson Chi-Square: 29.320 (4) *** Likelihood Ratio: 28.675 (4) *** Linear-by-Linear Association: 22.628 (1) *** N of Valid Cases: 1747
Disagree	27 (17.4) 11.6	84 (90.3) 7.0	20 (23.3) 6.4	131 7.5	
Not sure	90 (66.7) 38.8	343 (346.0) 28.5	69 (89.4) 22.2	502 28.7	

*** Statistically significant at *p* < 0.01.

**Table 6 ijerph-17-08190-t006:** Knowledge in waste separation and waste separation behaviour.

Knowledge in Related Field	Waste Separation Behaviour Top Figures: Observed (expected) Counts Bottom Figures: % within Column			Total	Chi-Square Tests Figures in Brackets: Degree of Freedom
	Never	Sometimes	Always		
Question: Are you familiar with the waste separation standards in Zhengzhou?					
Familiar	9 (12.7) 3.8	56 (66.2) 4.7	31 (17.1) 10.0	96 5.5	Pearson Chi-Square: 46.691 (4) *** Likelihood Ratio: 48.322 (4) *** Linear-by-Linear Association: 38.847 (1) *** N of Valid Cases: 1747
Unfamiliar	202 (170.8) 87.1	889 (886.3) 73.8	195 (228.9) 62.7	1286 73.6	
Not sure	21 (48.5) 9.1	259 (251.6) 21.5	85 (65.0) 27.3	365 20.9	

*** Statistically significant at *p* < 0.01.

**Table 7 ijerph-17-08190-t007:** Attitude towards waste separation and waste separation behaviour.

Attitude	Waste Separation Behaviour Top Figures: Observed (expected) Counts Bottom Figures: % within Column			Total	Chi-Square Tests Figures in Brackets: Degree of Freedom
	Never	Sometimes	Always		
Statement 1: Waste separation can improve our living environment and public health					
Agree	200 (215.4) 86.2	1126 (1117.9) 93.5	296 (288.7) 95.2	1622 92.8	Pearson Chi-Square: 25.294 (4) *** Likelihood Ratio: 20.155 (4) *** Fisher’s Exact Test: 20.680 *** Linear-by-Linear Association: 15.049 (1) *** N of Valid Cases: 1747 Note: 2 cells (22.2%) have expected count less than 5.
Disagree	11 (3.7) 4.7	13 (19.3) 1.1	4 (5.0) 1.3	28 1.6	
Not sure	21 (12.9) 9.1	65 (66.9) 5.4	11 (17.3) 3.5	97 5.6	
Statement 2: Universities should act as role models in waste separation					
Agree	188 (208.9) 81.0	1092 (1084.1) 90.7	293 (280.0) 94.2	1573 90.0	Pearson Chi-Square: 41.162 (4) *** Likelihood Ratio: 35.150 (4) *** Linear-by-Linear Association: 21.802 (1) *** N of Valid Cases: 1747
Disagree	16 (5.3) 6.9	17 (27.6) 1.4	7 (7.1) 2.3	40 2.3	
Not sure	28 (17.8) 12.1	95 (92.4) 7.9	11 (23.9) 3.5	134 7.7	
Statement 3: Waste separation helps reduce our carbon footprints and is good for the environment					
Agree	184 (195.5) 79.3	1024 (1014.5) 85.1	264 (262.0) 84.9	1472 84.3	Pearson Chi-Square: 14.830 (4) *** Likelihood Ratio: 12.801 (4) *** Linear-by-Linear Association: 4.388 (1) *** N of Valid Cases: 1747
Disagree	29 (15.8) 12.5	68 (82.0) 5.6	22 (21.2) 7.1	119 6.8	
Not sure	19 (20.7) 8.2	112 (107.5) 9.3	25 (27.8) 8.0	156 8.9	

*** Statistically significant at *p* < 0.01.

**Table 8 ijerph-17-08190-t008:** Subjective norms and waste separation behaviour.

Subjective Norms	Waste Separation Behaviour Top Figures: Observed (expected) Counts Bottom Figures: % within Column			Total	Chi-Square Tests Figures in Brackets: Degree of Freedom
	Never	Sometimes	Always		
Statement 1: If people around me do waste separation, I would do it too					
Agree	194 (204.1) 83.6	1058 (1059.3) 87.9	285 (273.6) 91.7	1537 88.0	Pearson Chi-Square: 16.313 (4) *** Likelihood Ratio: 15.152 (4) *** Linear-by-Linear Association: 8.522 (1) *** N of Valid Cases: 1747
Disagree	16 (7.8) 6.9	33 (40.7) 2.7	10 (10.5) 3.2	59 3.4	
Not sure	22 (20.1) 9.5	113 (104.1) 9.4	16 (26.9) 5.1	151 8.6	
Statement 2: I feel guilty if I do not separate my waste or if I do not do it properly					
Agree	113 (151.4) 48.7	775 (785.7) 64.4	252 (202.9) 81.0	1140 65.3	Pearson Chi-Square: 75.819 (4) *** Likelihood Ratio: 73.740 (4) *** Linear-by-Linear Association: 67.995 (1) *** N of Valid Cases: 1747
Disagree	40 (18.5) 17.2	87 (95.8) 7.2	12 (24.7) 3.9	139 8.0	
Not sure	79 (62.1) 34.1	342 (322.5) 28.4	47 (83.3) 15.1	468 26.7	

*** Statistically significant at *p* < 0.01.

**Table 9 ijerph-17-08190-t009:** Reward and penalty system and waste separation behaviour.

Reward and Penalty System	Waste Separation Behaviour Top Figures: Observed (expected) Counts Bottom Figures: % within Column			Total	Chi-Square Tests Figures in Brackets: Degree of Freedom
	Never	Sometimes	Always		
Question 1: Without any reward and penalty measures, would you do waste separation?					
Yes	149 (184.3) 64.2	960 (956.6) 79.7	279 (247.1) 89.7	1388 79.5	Pearson Chi-Square: 53.056 (2) *** Likelihood Ratio: 52.197 (2) *** Linear-by-Linear Association: 51.298 (1) *** N of Valid Cases: 1747
No	83 (47.7) 35.8	244 (247.4) 20.3	32 (63.9) 10.3	359 20.5	
Question 2: If you would be penalized for not separating your garbage, would you do waste separation?					
Yes	212 (214.3) 91.4	1138 (1112.3) 94.5	264 (287.3) 84.9	1614 92.4	Pearson Chi-Square: 32.980 (2) *** Likelihood Ratio: 28.629 (2) *** Linear-by-Linear Association: 11.601 (1) *** N of Valid Cases: 1747
No	20 (17.7) 8.6	66 (91.7) 5.5	47 (23.7) 15.1	133 7.6	
Question 3: If you would be rewarded for separating your waste, would you do waste separation?					
Yes	203 (218.9) 87.5	1146 (1135.8) 95.2	299 (293.4) 96.1	1648 94.3	Pearson Chi-Square: 23.792 (2) *** Likelihood Ratio: 19.194 (2) *** Linear-by-Linear Association: 15.986 (1) *** N of Valid Cases: 1747
No	29 (13.1) 12.5	58 (68.2) 4.8	12 (17.6) 3.9	99 5.7	

*** Statistically significant at *p* < 0.01.

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
