# Peer review of "Waste Separation Behaviour of College Students under a Mandatory Policy in China: A Case Study of Zhengzhou City"

_ijerph, 2020, doi:10.3390/ijerph17218190_

Round 1
Reviewer 1 Report
The paper addresses an essential element of any solid waste management system (SWM) – people’s behaviour. The paper is well written.
Having said that, the paper lacks sufficient depth of analysis and policy implications. While numerous statistical analyses of the survey data are carried out, the interpretation of their results in the context of SWM is rather shallow. Similarly, while the authors have done a fair literature search and generally provide an adequate comparison of their findings with literature, there is a lack of comprehensive understanding of the way SWM systems are put in place and function in practice. There is also lack of comprehensive familiarity with theories of human behaviour, which has consequences for the choice of answers offered to respondents.
Also, particularly in its concluding sections 4 and 5, the paper is a summation of praises of new measures, rather than a critical consideration of the research findings.
I provide detailed comments in a separate document so that the paper can be revised in terms of its scientific and policymaking relevance, which the authors seek to achieve.

Reviewer 2 Report
Line 12-30. The Abstract is far from the research title. I don’t know how discovered problem ‘it was found that college students do have a basic knowledge of waste separation and most are familiar with where kitchen waste should be placed, but they have problems categorizing some recyclables such as glass, hazardous waste such as light tubes and other waste such as cigarette ends and napkins’ might be caused by the ‘mandatory regulation’. Your argument does not support the research objective. Reformulate it!
Line 33-172. The section is so long and more a literature review rather than an introduction. Also, you need to emphasize your intention more clearly. The current introduction comprised a wide range of influential factors that could be summarized and specified toward the research objectives.
Line 149-151. You already reject this argument on subsequent lines 162-165! This inconsistency can be seen also in other parts of your article. Review the whole manuscript and reformulated them.
Line 167-168. What are these mandatory regulations? You didn’t explain it here and I missed it seriously.
Line 179. Which questions are open-question? They all as far as I notice are closed since your participants were able to select between two or three options i.e. never, sometimes, and always.
Line 181-189. Where is/are the questions that address constraints or opportunities that provided and forced by mandatory regulations? Should we know how these regulations affect residents’ behavior? Or you just assumed that these regulations spontaneously influenced your case study?
Line 190. This is magnificent but please elaborate it: How you manage the time especially considering the COVID-19 situation that stroked China? How you convenient the participants to be part of the face-to-face interview in such a period and how six of you have done a 1689 questionnaire consists of 27 questions from 62 universities in just one month (January 2020).
Line 225. Check the number. 56% does not equal to 232 persons.
Line 231. Use the same scale everywhere. Here are the percentage while previously was percentage and number.
Line 247. How this section related to the research objective and how does it connect to mandatory regulation? If it is not the case then why it was discussed here?
This will be the same case for other sections, how mandatory regulation has changed residents’ attitude? (Section 3.3.2), or within section 3.3.3 and so on. I think you are agreed with me that your discussion within the whole manuscript, shown the effect of other factors rather than mandatory regulation.
Line 417-418. The research has done by dai et al (2017) is your main point of argument but you didn’t explain that whether the time gap between these two studies, the number of case studies, the condition and…etc., between your research and the previous one, can be ignored and has no effect on recent data captured by your research.
Line 470-474. You need to elaborate on how in such a short period (the regulation implemented in 2019 and your research initiated in Nov 2019) the regulations were able to change residents’ behavior? And statistically demonstrates its effects.
Line 497-501. How mandatory regulations are involved with this suggestion?
Line 502-510. Do you believe it will work in the long-term?
Line 512,513. There are typos, SWM?
Generally, I think what you have done with your team and your efforts are rewarding indeed. Yet, you need to revise the manuscript in many ways to orient toward the title or revise the title. Your research shows the effect of recycling knowledge, attitude, collecting waste facilities, perceived convenience is indeed impactful which is also mentioned by former studies. The role of mandatory regulation and how it was impactful, however, is missing information that leaves your main argument rather than vague. For the revised paper I expect you to clarify the implemented regulations and how these regulations contributed to changing your research outcomes? How these regulations have changed residents' approach to whether sort the generated waste or not? Without having this information at hand it would be impossible to criticize them or improve them. Good luck!
Reviewer 3 Report
Waste separation behavior is a problem worthy of attention. This study focuses on waste separation behaviour of college students under mandatory policy in China. The topic is hot, however, the content of the article and the conclusion of the study is not interesting and novel. There are some obvious defects for the paper, it has not yet reached the requirement of publication. The concerns are provided below: 1. The first section is too long, and the Introduction pays more attention to the research background. Although this content is necessary, the history of policies and studies should not be elaborated too much. In addition, the research significance and necessity are not clear in this part. 2. Why only choose college students to do? What is the practical significance of only studying this group? 3. A questionnaire was used to collect the relevant data. But I did not see the reliability and validity of the scale test, which can not prove the reliability and credibility of the data, it is impossible to know the scientific conclusions. Is it just that data on this group is easy to collect? 4. The subjectivity of the questionnaire survey is strong, the research process of the research data is relatively vague, can not see how the author is to ensure that the research data is true. 5. The structure of the paper is incomplete, there is no “Discussion” in the end of data analysis. The discussion usually contains a synthesis of the findings, the practical implications, the theoretical implications, the strengths and limitations of the research, and the future research directions. However, some contents are missing in this paper. 6. The conclusions of the study are not innovative. In particular, the content and conclusions of the study bear little relation to the "mandatory policy" in the Title and Instruction.Author Response
Please see the attachment.

Reviewer 4 Report
The paper entitled “Waste separation behaviour of college students under mandatory policy in China: A case study of Zhengzhou City” discusses an overall interesting issue, that becomes even more relevant considering the recent introduction of mandatory waste separation measures by the city of Zhengzhou. However, I do not believe it is suitable for publication in its current form, and minor changes are required.
The literature review provides sufficient background and displays adequate knowledge about the topic. References to main previous research are provided.
The research design is appropriate. However, I would strongly suggest adding a graphic model explaining the relations under investigation. Several variables are employed in the questionnaire and a visual representation could really help readers to gain a better understanding.
The method section must be improved, since no reference for scales and items used in the questionnaire is provided.
Results are clearly presented, but as suggested before, a graphic representation would undoubtedly help. Moreover, I do not consider appropriate the comparison of the current waste separation behavior with the past situation. Previous studies, and in particular Dai et al. 2017, do not adopt the same unit of analysis nor the same research tools, and therefore the scientific soundness of this section is not adequate.
Main conclusions are supported by the results of the study, except for the comparison before and after the introduction of mandatory measures, as I mentioned before.
English language and style are fine. Tables and figures layout could be improved.
Reviewer 5 Report
Review comments on ijerph-945088
Major revision is required. The topic discussed is interesting to me. However, in order to warrant publication, the following improvements or clarifications are to be made:
- There is no mention of ethical approval obtained for research involving human objects. Authors please provide the approval code and or approval body.
- The aim of this manuscript is said to “address these gaps in knowledge…” (line 170). However, the study only covers University students and non-probability sampling was used. I wonder how representative the findings (derived from non-probabiity sampling) revealed in this University student sampling pool is given that given that some of the gaps mentioned were national wide questions and do not confine to just one small sector of a society.
- Why was non-probability and not probability sampling method chosen?
- There were two ways to obtain the samples, are there any measures to prevent duplication of entries?
- Why is Fisher’s exact test result quoted for the discussion on income and waste separation behavior (line 273)? I took the trouble to look up from the manual of SPSS and I find this: “For 2 × 2 tables, Fisher's exact test is computed when a table that does not result from missing rows or columns in a larger table has a cell with an expected frequency of less than 5.” First the last section of Table 2 is not a 2x2 table and second, there were 66.7% of cells with expected count less than 5. If I were you, I will co-code of data to get rid of the problematic cells before I will perform any chi-sq. test. This applies to the cross-tab between waste separation behavior and the attitude “waste separation can improve our living environment and public health”.
- There is inadequate discussion on the exact requirement of the Zhengzhou Measures. Any preliminary implementation evaluation/effectiveness known already?
- Being in the University for many years, I am well aware that the importance of studying the environmental awareness, attitude and behavour of university students has been very much downplayed by the academic sector. But this is not to say that one can publish just about any environmental attitude studies about college students. How is the waste separation facilities and programmes like in those sampled Universities? Was waste separation mandated at those Universities? Why not, if the city has already mandated source separation of waste? These are necessary background information for the readers to understand why the college students behave or think in the way revealed by the study.
- Please explain how you obtain the data for Figure 2. Did you conduct any observation/auditing of separated waste?
- For the reward and penalty questions, I do not think that the results can be used. There are just two values for the respondents to choose: yes or no. However, I think in reality the situation is more complex. People may be committed to waste separation because they see it as their duty to do it. If a reward is given, the agents may regard their waste sorting behavior demoted and no longer want to carry it out. I suggest not using the data from this badly designed question.
- The format of Table 9 is not reading friendly.
Round 2
Reviewer 1 Report
The paper has been remarkably improved. Now it is a valuable scientific contribution and a pleasure to read.
A few final comments are given in a separate document. They concern improvements of readability.

Reviewer 2 Report
The manuscript has improved indeed. Still, there two issues that I need you to fix it before publishing and within the final version;
1.Within Figure 1, the relation between variables needed to clarify if one is the result of the other variable and if it is the case where it should be located? e.g. if the box comprises 'waste separation behavior under mandatory policy' is resulted from the previous box then logically should be placed below than the box, not on the upper side of the box. Careful about using arrows!
2. You clarified the interview procedure in the present version of your manuscript and I don't believe that it would be necessary to have pictures here (Figure 2) as an improvement of your statement. So if this issue was not of concern to other reviewers just remove it.
Thank you for your good job and good luck!

Reviewer 3 Report
1. I still believe that the sample selection of college students cannot represent the residents of Zhengzhou, and at the same time, the college students of Zhengzhou cannot replace the entire Chinese residents.
2. In Figure 1, I did not find the role of mandatory separation policy in the model. If the mandatory separation policy is removed, it is also suitable. So, what is the theoretical significance of this model?
3. It was mentioned that:” But little is known about the factors that influence the waste separation behaviour of college students, especially in countries like China which are undergoing extensive and rapid economic expansion and urbanization.” In fact, many studies are studying the problem of Chinese residents’ waste separation behavior. College students are also Chinese residents. Such as:
Dongliang Z , Guangqing H , Xiaoling Y , et al. Residents' Waste Separation Behaviors at the Source: Using SEM with the Theory of Planned Behavior in Guangzhou, China. International Journal of Environmental Research & Public Health, 2015, 12(8):9475-9491.
Chen F , Chen H , Wu M , et al. Research on the Driving Mechanism of Waste Separation Behavior: Based on Qualitative Analysis of Chinese Urban Residents[J]. International Journal of Environmental Research and Public Health, 2019, 16(10):1859.
Wang R, Sun C, Cheng J, et al. Impact of ecological civilization education in universities on the students’ waste classification behavior - evidences from 152 universities in China. Journal of Arid Land Resources and Environment, 2020.
The author does not fully understand the relevant theoretical research.
